# Mode of birth and risk of infection-related hospitalisation in childhood: A population cohort study of 7.17 million births from 4 high-income countries

Jessica E. Miller[1,2]*, Raphael Goldacre[3], Hannah C. Moore[4], Justin Zeltzer[5], Marian Knight[6], Carole Morris[7], Sian Nowell[7], Rachael Wood[7], Kim W. Carter[8], Parveen Fathima[4], Nicholas de Klerk[4], Tobias Strunk[9], Jiong Li[10], Natasha Nassar[5], Lars H. Pedersen[11,12,13☯], David P. Burgner[1,2,14☯]

1 Murdoch Children's Research Institute, The Royal Children's Hospital, Parkville, Victoria, Australia, 2 University of Melbourne, Department of Paediatrics, Parkville, Victoria, Australia, 3 University of Oxford, Big Data Institute, Nuffield Department of Population Health, Oxford, United Kingdom, 4 Wesfarmers Centre of Vaccines and Infectious Diseases, Telethon Kids Institute, University of Western Australia, Perth, Australia, 5 University of Sydney, Child Population and Translational Health Research, Children's Hospital at Westmead Clinical School, Sydney, Australia, 6 University of Oxford, National Perinatal Epidemiology Unit, Nuffield Department of Population Health, Oxford, United Kingdom, 7 NHS National Services Scotland, Information Services Division, Edinburgh, United Kingdom, 8 Telethon Kids Institute, University of Western Australia, Perth, Australia, 9 Centre for Neonatal Research and Education, University of Western Australia, Perth, Australia, 10 Aarhus University, Department of Clinical Epidemiology, Aarhus, Denmark, 11 Aarhus University, Department of Clinical Medicine, Aarhus, Denmark, 12 Aarhus University, Department of Biomedicine, Aarhus, Denmark, 13 Aarhus University Hospital, Department of Obstetrics and Gynaecology, Aarhus, Denmark, 14 Monash University, Department of Paediatrics, Clayton, Victoria, Australia

☯ These authors contributed equally to this work.
* jessica.miller@mcri.edu.au

**Data Availability Statement:** Regulations regarding access to jurisdictional data vary. Please visit the following websites for information

## Abstract

### Background

The proportion of births via cesarean section (CS) varies worldwide and in many countries exceeds WHO-recommended rates. Long-term health outcomes for children born by CS are poorly understood, but limited data suggest that CS is associated with increased infection-related hospitalisation. We investigated the relationship between mode of birth and childhood infection-related hospitalisation in high-income countries with varying CS rates.

### Methods and findings

We conducted a multicountry population-based cohort study of all recorded singleton live births from January 1, 1996 to December 31, 2015 using record-linked birth and hospitalisation data from Denmark, Scotland, England, and Australia (New South Wales and Western Australia). Birth years within the date range varied by site, but data were available from at least 2001 to 2010 for each site. Mode of birth was categorised as vaginal or CS (emergency/elective). Infection-related hospitalisations (overall and by clinical type) occurring after the birth-related discharge date were identified in children until 5 years of age by primary/secondary *International Classification of Diseases*, 10th Revision (ICD-10) diagnosis

regarding access to the same data used in this project: Denmark - Statistics Denmark website: https://www.dst.dk/en/TilSalg/skraeddersyede-loesninger or contact forskningsservice@dst.dk England - NHS Digital website: www.digital.nhs.uk Scotland - Public Health Scotland website: https://publichealthscotland.scot/ or contact phs.edris@nhs.net New South Wales, Australia - NSW Centre for Health Record Linkage website: https://www.cherel.org.au/ Western Australia - Data Linkage Western Australia website: https://www.datalinkage-wa.org.au/.

**Funding:** NdK, KWC, and DPB received funding from National Health and Medical Research Council project grants www.nhmrc.gov.au (GTN1065494: NdK, KWC, DPB), (GTN1045668: HCM, NdK), Fellowship (1034254: HCM), and Senior Research Fellowship (GTN1064629: DPB); JEM received funding from the DHB Foundation; LHP received funding from Health Research Fund of Central Denmark Region; JL received funding from the Novo Nordisk Foundation www.novonordisk.com (NNF18OC0052029), and the Danish Council for Independent Research https://dff.dk/en (DFF-6110-00019); NN received funding from Financial Markets Foundation for Children www.foundationforchildren.com.au; TS received funding from Raine Foundation Clinician Research Fellowship http://rainefoundation.org.au; RG and MK received funding from Public Health England www.gov.uk/government/organisations/public-health-england, the Li Ka Shing Foundation www.lksf.org, the Robertson Foundation www.robertsonfoundation.org, the Medical Research Council https://mrc.ukri.org, British Heart Foundation www.bhf.org.uk, and the NIHR Oxford Biomedical Research Centre https://oxfordbrc.nihr.ac.uk. The funders had no role in study design, data collection and analysis, decision to publish, or preparation of the manuscript.

**Competing interests:** The authors have declared that no competing interests exist.

**Abbreviations:** CI, confidence interval; CS, cesarean section; HR, hazard ratio; ICD-10, *International Classification of Diseases*, 10th Revision; NHS, National Health Service; NICE, National Institute for Health and Care Excellence; RECORD, REporting of studies Conducted using Observational Routinely-collected Data.

codes. Analysis used Cox regression models, adjusting for maternal factors, birth parameters, and socioeconomic status, with results pooled using meta-analysis. In total, 7,174,787 live recorded births were included. Of these, 1,681,966 (23%, range by jurisdiction 17%–29%) were by CS, of which 727,755 (43%, range 38%–57%) were elective. A total of 1,502,537 offspring (21%) had at least 1 infection-related hospitalisation. Compared to vaginally born children, risk of infection was greater among CS-born children (hazard ratio (HR) from random effects model, HR 1.10, 95% confidence interval (CI) 1.09–1.12, $p < 0.001$). The risk was higher following both elective (HR 1.13, 95% CI 1.12–1.13, $p < 0.001$) and emergency CS (HR 1.09, 95% CI 1.06–1.12, $p < 0.001$). Increased risks persisted to 5 years and were highest for respiratory, gastrointestinal, and viral infections. Findings were comparable in prespecified subanalyses of children born to mothers at low obstetric risk and unchanged in sensitivity analyses. Limitations include site-specific and longitudinal variations in clinical practice and in the definition and availability of some data. Data on postnatal factors were not available.

## Conclusions

In this study, we observed a consistent association between birth by CS and infection-related hospitalisation in early childhood. Notwithstanding the limitations of observational data, the associations may reflect differences in early microbial exposure by mode of birth, which should be investigated by mechanistic studies. If our findings are confirmed, they could inform efforts to reduce elective CS rates that are not clinically indicated.

## Author summary

### Why was this study done?

- Health outcomes beyond the neonatal period for children born by cesarean section (CS) are not well understood.

- CS may be associated with an increased risk of severe childhood infection requiring hospitalisation, but data are limited.

- Whether CS is associated with increased risk of overall infection or only certain types of infection and whether the risk differs for emergency versus elective CS is unclear.

### What did the researchers do and find?

- Using total population birth and hospitalisation data from Denmark, Scotland, England, and Australia (New South Wales and Western Australia), we followed all recorded singleton live births from January 1, 1996 to December 31, 2015, for up to 5 years to determine whether children were admitted to hospital with an infection.

- We estimated risk of overall and clinical type of infection by mode of birth, vaginal, or CS (emergency/elective).

- Among 7.17 million births, children born by elective CS, compared to vaginally born children, had a 13% increased risk for an infection-related hospitalisation and emergency CS-born children had a 9% increased risk.

- Increased risks persisted to 5 years of age and were highest for respiratory, gastrointestinal, and other viral infections.

## What do these findings mean?

- In our large multinational study, we observed a consistent association between birth by CS and infection-related hospitalisation in early childhood.

- Limitations include site-specific and longitudinal variations in clinical practice.

- The associations may reflect differences in early microbial exposure by mode of birth, which should be investigated by mechanistic studies.

- These findings may contribute to the global effort to reduce the rates of elective CS that are not medically indicated.

## Introduction

Cesarean section (CS) may be a lifesaving intervention for women and babies and is the most common major surgical procedure in many countries. Since 2000, the global proportion of CS births has nearly doubled, but this increase may not be medically justified [1]. An estimated 6.2 million nonmedically indicated CSs are performed annually worldwide [2]. Recent estimates of the proportion of births by CS vary markedly by region: 4% in sub-Saharan Africa, 20% to 30% in Europe, United States, and Australia, over 40% in some regions of China, and over 70% in some private hospitals in Vietnam and Brazil [1,3–5].

CS has short- and long-term health implications for both mother and child; the increasing rates warrant population-level analyses of potential risks. A study in low- and middle-income countries reported disproportionately high rates of maternal and perinatal death following CS [6]. Many suggested long-term adverse outcomes in CS-born children, including increased risk of asthma, allergy, juvenile idiopathic arthritis, and inflammatory bowel disease, relate to altered immune development, and risk may vary depending on CS type (elective or emergency) [2,7,8]. Differences in the newborn microbiome by mode of birth determine early immune responses [9,10] and may influence the risk of immune-related outcomes, including infection.

There are limited data on the relationship between mode of birth and common childhood infections beyond the neonatal period. An increased risk of specific infection-related hospitalisations, mainly lower respiratory tract and gastrointestinal infections, has been associated with CS [11,12]. An Australian study of term singleton births ($n$ = 212,068) found an 11% and 20% increased risk of hospitalisation with bronchiolitis in children aged <12 months and 12 to 23 months, respectively [13], and a Danish study ($n$ = 750,569) reported similarly increased risk for lower respiratory tract infection [7]. CS has also been associated with increased risk of childhood gastroenteritis [14]. A recent Israeli study among uncomplicated pregnancies and births ($n$ = 138,910) estimated a 10% and 23% increased risk of hospitalisations with infection

up to age 18 years among term-only and preterm-only elective CS births, respectively [15]. These studies are from single jurisdictions where population-specific characteristics and obstetric practice may have unknown effects. It is unclear if CS is associated with increased risk of overall infection-related hospitalisation or only certain infection types, whether risk differs for emergency versus elective CS, and if the associations may partly represent confounding by indication.

We investigated the association between mode of birth and infection-related hospitalisation in 5 populations from 4 countries with varying CS rates. Follow-up was until 5 years of age, the period of greatest infection burden [16,17]. We hypothesised that the risk of infection-related hospitalisation would be highest for (1) children born by elective CS who are not exposed to maternal vaginal microbiome during delivery; and (2) for infections of sites such as respiratory tract and gastrointestinal where direct inoculation of maternal vaginal microbiome may contribute to optimal early mucosal immune responses.

## Methods

### Study population

Data were from population-level databases in Denmark, Scotland, England, and Australia (New South Wales and Western Australia) and comprised linked administrative (including birth and death) and hospital data [18–23]. All recorded live-born singletons were identified from each site (S1 Fig). Birth years ranged from January 1, 1996 to December 31, 2015, with data from at least 2001 to 2010 available from each site. Children with congenital malformations (*International Classification of Diseases*, 10th Revision (ICD-10) codes Q00–Q99) were excluded as some conditions may be associated with mode of birth and postnatally with increased infection risk.

### Exposure and outcomes

Mode of birth was categorised as vaginal or CS and by type (emergency or elective) based on recorded data in the birth databases (S1 Table). Children were classified as having an infection-related hospitalisation if they had an inpatient hospital admission with 1 or more primary or secondary infection-related discharge codes (ICD-10), at least 1 day after the birth-related discharge and were less than 5 years old at discharge. Date of onset was defined as the first recorded day of contact with the hospital when patients were hospitalised. Rehospitalisation for infection within 7 days was considered as a single admission. ICD-10–coded infections were classified a priori into 7 clinical groups: invasive bacterial, skin and soft tissue, genitourinary, upper respiratory tract, lower respiratory tract, viral infections, and gastrointestinal, as previously described [16].

### Additional variables

Maternal factors and birth parameters are shown in Table 1. Gestational age range was 24 to 43 weeks, with the exception of England where gestational age data for <30 and >42 weeks were not deemed reliable and excluded [24]. Birth weight range was 500 to 5,500 grams for all sites. Measures of socioeconomic status for the year closest to birth were based on either area level deprivation (quintiles) for Scotland, England, and Australia, or parents' highest level of completed education ["low" (high school education or less), "middle" (college or vocational training), or "high" (graduate-level education)] for Denmark. Data on hypertensive disorders or diabetes mellitus during pregnancy and labour onset were from birth and hospital data collections (data on labour onset were unavailable for England and Scotland) (S1 Table).

**Table 1. Characteristics of the study populations.**

| | Denmark | | | Scotland | | | England | | | New South Wales | | | Western Australia | | |
|---|---|---|---|---|---|---|---|---|---|---|---|---|---|---|---|
| Birth data: | 1997–2010 | | | 2001–2015 | | | 1 April 1998–31 March 2012 | | | 2001–2012 | | | 1996–2012 | | |
| Hospital data: | 1997–2015 | | | 2001–2016 | | | 1 April 1998–31 March 2012 | | | 2001–2012 | | | 1996–June 30 2013 | | |
| | n = 783,082 | | | n = 719,625 | | | n = 4,289,829 | | | n = 945309 | | | n = 436942 | | |
| Characteristic | Vaginal Births N (%) | Emergency Caesarean Sections N (%) | Elective Caesarean Sections N (%) | Vaginal Births N (%) | Emergency Caesarean Sections N (%) | Elective Caesarean Sections N (%) | Vaginal Births N (%) | Emergency Caesarean Sections N (%) | Elective Caesarean Sections N (%) | Vaginal Births N (%) | Emergency Caesarean Sections N (%) | Elective Caesarean Sections N (%) | Vaginal Births N (%) | Emergency Caesarean Sections N (%) | Elective Caesarean Sections N (%) |
| | 649955 (83.0) | 76560 (9.8) | 56567 (7.2) | 537816 (74.7) | 106968 (14.9) | 74841 (10.4) | 3312535 (77.2) | 601430 (14) | 375864 (8.8) | 683940 (72.4) | 111320 (11.8) | 150049 (15.9) | 308575 (70.6) | 57933 (13.3) | 70434 (16.1) |
| **Sex** | | | | | | | | | | | | | | | |
| Female | 325023 (50.0) | 34981 (45.7) | 29018 (51.3) | 269402 (50.1) | 48393 (45.2) | 37888 (50.6) | 1660439 (50.1) | 276929 (46) | 188941 (50.3) | 340554 (49.8) | 50059 (45) | 73938 (49.3) | 154390 (50) | 26705 (46.1) | 35123 (49.9) |
| Male | 324587 (49.9) | 41555 (54.3) | 27526 (48.7) | 268390 (49.9) | 58573 (54.8) | 36953 (49.4) | 1604732 (48.4) | 315509 (52.5) | 182110 (48.5) | 343195 (50.2) | 61201 (55) | 75932 (50.6) | 154185 (50) | 31228 (53.9) | 35311 (50.1) |
| Missing | 345 (0) | 24 (0) | 23 (0) | 24 (0) | 2 (0) | 0 (0) | 47364 (1.4) | 8992 (1.5) | 4813 (1.3) | 191 (0) | 60 (0.1) | 179 (0.1) | 0 (0) | 0 (0) | 179 (0.1) |
| **Gestational Age (weeks)** | | | | | | | | | | | | | | | |
| <28 | 295 (0) | 335 (0.4) | 56 (0.1) | 484 (0.1) | 324 (0.3) | 9 (0) | N/A[a] | N/A[a] | N/A[a] | 487 (0.1) | 97 (0.1) | 158 (0.1) | 787 (0.3) | 469 (0.8) | 28 (0) |
| 28–<32 | 1179 (0.2) | 1484 (1.9) | 295 (0.5) | 1511 (0.3) | 1977 (1.8) | 169 (0.2) | 6613 (0.2) | 7289 (1.2) | 1003 (0.3) | 1380 (0.2) | 488 (0.4) | 1240 (0.8) | 1022 (0.3) | 1125 (1.9) | 92 (0.1) |
| 32–<34 | 1983 (0.3) | 1819 (2.4) | 529 (0.9) | 2275 (0.4) | 2251 (2.1) | 374 (0.5) | 15324 (0.5) | 13007 (2.2) | 2253 (0.6) | 2708 (0.4) | 797 (0.7) | 1871 (1.2) | 1519 (0.5) | 1219 (2.1) | 215 (0.3) |
| 34–<36 | 9073 (1.4) | 3597 (4.7) | 1337 (2.4) | 7820 (1.5) | 4357 (4.1) | 1190 (1.6) | 50506 (1.5) | 25066 (4.2) | 6437 (1.7) | 9522 (1.4) | 2410 (2.2) | 4127 (2.8) | 5820 (1.9) | 2961 (5.1) | 903 (1.3) |
| 36–<38 | 44477 (6.8) | 8197 (10.7) | 9587 (16.9) | 35243 (6.6) | 9927 (9.3) | 5918 (7.9) | 228737 (6.9) | 60716 (10.1) | 38829 (10.3) | 47256 (6.9) | 9744 (8.8) | 14902 (9.9) | 28795 (9.3) | 8961 (15.5) | 10853 (15.4) |
| 38–40 | 243488 (37.5) | 21756 (28.4) | 39965 (70.6) | 347744 (64.7) | 53227 (49.8) | 63219 (84.5) | 2177278 (65.7) | 313586 (52.1) | 305492 (81.3) | 495779 (72.5) | 69375 (62.3) | 122162 (81.4) | 495779 (72.5) | 228010 (73.9) | 33799 (58.3) |
| >40 | 342815 (52.7) | 38839 (50.7) | 4643 (8.2) | 141889 (26.4) | 34789 (32.5) | 3894 (5.2) | 834077 (25.2) | 181766 (30.2) | 21850 (5.8) | 126808 (18.5) | 28409 (25.5) | 5589 (3.7) | 42622 (13.8) | 9399 (16.2) | 1520 (2.2) |
| Missing | 6645 (1.0) | 533 (0.7) | 155 (0.3) | 850 (0.2) | 116 (0.1) | 68 (0.1) | 0 (0) | 0 (0) | 0 (0) | 0 (0) | 0 (0) | 0 (0) | 0 (0) | 0 (0) | 0 (0) |
| **Birth Weight z-score (gestational age- and sex-specific)** | | | | | | | | | | | | | | | |
| ≤10 | 67430 (10.4) | 10043 (13.1) | 5192 (9.2) | 40263 (7.5) | 9668 (9) | 3359 (4.5) | 326211 (9.8) | 70472 (11.7) | 26193 (7) | 69452 (10.2) | 11689 (10.5) | 12576 (8.4) | 30176 (9.8) | 6632 (11.5) | 4820 (6.8) |
| >10–25 | 98935 (15.2) | 10155 (13.3) | 7619 (13.5) | 72006 (13.4) | 12484 (11.7) | 6156 (8.2) | 509247 (15.4) | 82370 (13.7) | 43421 (11.6) | 108537 (15.9) | 14996 (13.5) | 19273 (12.8) | 47252 (15.3) | 8003 (13.8) | 8851 (12.6) |
| >25–75 | 327933 (50.4) | 34249 (44.7) | 27870 (49.3) | 299711 (52) | 49137 (45.9) | 34164 (45.6) | 1633823 (49.3) | 265276 (44.1) | 172861 (46) | 347866 (50.9) | 52338 (47) | 71741 (47.8) | 156938 (50.9) | 27532 (47.5) | 34800 (49.4) |
| >75–90 | 95963 (14.8) | 11987 (15.7) | 8929 (15.8) | 85951 (16) | 18112 (16.9) | 15262 (20.4) | 478383 (14.4) | 90690 (15.1) | 65541 (17.4) | 97346 (14.2) | 17633 (15.8) | 25272 (16.8) | 45921 (14.9) | 8849 (15.3) | 12302 (17.5) |
| >90 | 53574 (8.2) | 9553 (12.5) | 6748 (15.8) | 58303 (10.8) | 17155 (16) | 15721 (21) | 317507 (9.6) | 83630 (13.9) | 63035 (16.8) | 60544 (8.9) | 14637 (13.1) | 21128 (14.1) | 28286 (9.2) | 6915 (11.9) | 9661 (13.7) |
| Missing | 6120 (0.9) | 573 (0.7) | 209 (0.4) | 1582 (0.3) | 412 (0.4) | 179 (0.2) | 47364 (1.4) | 8992 (1.5) | 4813 (1.3) | 195 (0) | 27 (0) | 59 (0) | 2 (0) | 2 (0) | 0 (0) |
| **Smoked during Pregnancy** | | | | | | | | | | | | | | | |
| No | 510812 (78.6) | 59175 (77.3) | 45092 (79.7) | 383784 (71.4) | 79590 (74.4) | 57854 (77.3) | N/A | N/A | N/A | 574274 (84) | 96463 (86.7) | 131672 (87.8) | 218730 (70.9) | 44590 (77) | 56363 (80) |
| Yes | 116671 (17.9) | 13949 (18.2) | 9116 (16.1) | 113754 (21.2) | 18755 (17.5) | 11428 (15.3) | N/A | N/A | N/A | 108012 (15.8) | 14587 (13.1) | 17788 (11.9) | 50295 (16.3) | 8255 (14.3) | 7962 (11.3) |
| Missing | 22472 (3.5) | 3436 (4.5) | 2359 (4.2) | 40278 (7.5) | 8623 (8.1) | 5559 (7.4) | N/A | N/A | N/A | 1654 (0.2) | 270 (0.2) | 589 (0.4) | 39550 (12.8) | 5088 (8.8) | 6109 (8.7) |
| **Maternal Age at Birth (years)** | | | | | | | | | | | | | | | |
| <20 | 10901 (1.7) | 974 (1.3) | 260 (0.5) | 41907 (7.8) | 5763 (5.4) | 1076 (1.4) | 248795 (7.5) | 29418 (4.9) | 7172 (1.9) | 29858 (4.4) | 3683 (3.3) | 1638 (1.1) | 19617 (6.4) | 2722 (4.8) | 859 (1.2) |
| 20–<25 | 80749 (12.4) | 8242 (10.8) | 3363 (5.9) | 107794 (20) | 16938 (15.8) | 7231 (9.7) | 684618 (20.7) | 95151 (15.8) | 38297 (10.2) | 108375 (15.8) | 13851 (12.4) | 10541 (7) | 56919 (18.5) | 8594 (14.8) | 5624 (8) |
| 25–<30 | 227374 (35.0) | 25756 (33.6) | 14464 (25.6) | 146545 (27.2) | 27642 (25.8) | 16657 (22.3) | 929914 (28.1) | 159301 (26.5) | 84759 (22.6) | 199146 (29.1) | 30594 (27.5) | 31637 (21.1) | 93808 (30.4) | 16524 (28.5) | 16509 (23.4) |
| 30–<35 | 229139 (35.3) | 26874 (35.1) | 22811 (40.3) | 151389 (28.1) | 32818 (30.7) | 25657 (34.3) | 904248 (27.3) | 182937 (30.4) | 126525 (33.7) | 218683 (32) | 37564 (33.7) | 55129 (36.7) | 218683 (32) | 91075 (29.5) | 18316 (31.6) |

*(Continued)*

**Table 1.** (Continued)

| | Denmark | | | Scotland | | | England | | | New South Wales | | | Western Australia | | |
|---|---|---|---|---|---|---|---|---|---|---|---|---|---|---|---|
| Birth data: | 1997–2010 | | | 2001–2015 | | | 1 April 1998–31 March 2012 | | | 2001–2012 | | | 1996–2012 | | |
| Hospital data: | 1997–2015 | | | 2001–2016 | | | 1 April 1998–31 March 2012 | | | 2001–2012 | | | 1996–June 30 2013 | | |
| | n = 783,082 | | | n = 719,625 | | | n = 4,289,829 | | | n = 945309 | | | n = 436942 | | |
| | Vaginal Births | Emergency Caesarean Sections | Elective Caesarean Sections | Vaginal Births | Emergency Caesarean Sections | Elective Caesarean Sections | Vaginal Births | Emergency Caesarean Sections | Elective Caesarean Sections | Vaginal Births | Emergency Caesarean Sections | Elective Caesarean Sections | Vaginal Births | Emergency Caesarean Sections | Elective Caesarean Sections |
| ≥35 | 101789 (15.7) | 14712 (19.2) | 15667 (27.7) | 90179 (16.8) | 23807 (22.3) | 24220 (32.4) | 530215 (16) | 131938 (21.9) | 117158 (31.2) | 127878 (18.7) | 25628 (23) | 51104 (34.1) | 47156 (15.3) | 11727 (20.2) | 14228 (20.2) |
| Missing | 3 (0) | 2 (0) | 2 (0) | 2 (0) | 2 (0) | 2 (0) | 14745 (0.4) | 2685 (0.4) | 1953 (0.5) | 0 (0) | 0 (0) | 0 (0) | 0 (0) | 0 (0) | 0 (0) |
| **Parity** | | | | | | | | | | | | | | | |
| 0 | 271252 (41.7) | 46869 (61.2) | 16827 (29.7) | 236663 (44) | 69895 (65.3) | 14867 (19.9) | 867675 (26.2) | 212067 (35.3) | 58549 (15.6) | 278461 (40.7) | 75026 (67.4) | 40348 (26.9) | 91541 (29.7) | 26173 (45.2) | 12913 (18.3) |
| 1 | 244158 (37.6) | 20588 (26.9) | 24796 (43.8) | 184603 (34.3) | 24042 (22.5) | 38372 (51.3) | 696846 (21) | 109325 (18.2) | 97193 (25.9) | 228007 (33.3) | 23444 (21.1) | 67604 (45.1) | 93801 (30.4) | 15649 (27) | 25327 (36) |
| 2 | 94045 (14.5) | 5830 (7.6) | 11213 (19.8) | 74244 (13.8) | 7669 (7.2) | 15099 (20.2) | 343707 (10.4) | 44529 (7.4) | 55164 (14.7) | 107933 (15.8) | 7737 (7) | 28542 (19) | 58477 (19) | 7637 (13.2) | 16278 (23.1) |
| ≥3 | 34937 (5.3) | 2532 (3.3) | 3468 (6.1) | 38556 (7.2) | 4408 (4.1) | 6017 (8) | 316577 (9.6) | 42196 (7) | 46499 (12.4) | 68659 (10) | 5002 (4.5) | 13223 (8.8) | 64756 (21) | 8474 (14.6) | 15916 (22.6) |
| Missing | 5563 (0.9) | 741 (1.0) | 263 (0.5) | 3750 (0.7) | 954 (0.9) | 486 (0.6) | 1087730 (32.8) | 193313 (32.1) | 118459 (31.5) | 880 (0.1) | 111 (0.1) | 332 (0.2) | 0 (0) | 0 (0) | 0 (0) |
| Area-level deprivation quintile (1 = most deprived, 5 = least deprived) (Denmark education level 1 = lowest, 3 = highest) | | | | | | | | | | | | | | | |
| 1 | 70046 (10.8) | 7631 (10.0) | 4611 (8.2) | 140224 (26.1) | 26498 (24.8) | 16781 (22.4) | 989367 (29.9) | 168767 (28.1) | 89835 (23.9) | 127642 (18.7) | 17047 (15.3) | 21533 (14.4) | 47475 (15.4) | 7763 (13.4) | 7285 (10.3) |
| 2 | 321160 (49.4) | 39009 (50.9) | 27816 (49.2) | 113160 (21) | 22369 (20.9) | 14480 (19.3) | 721468 (21.8) | 132322 (22) | 76356 (20.3) | 112553 (16.5) | 16344 (14.7) | 21864 (14.6) | 61189 (19.8) | 10978 (19) | 11265 (16) |
| 3 | 252631 (38.9) | 29180 (38.1) | 23777 (42.0) | 100199 (18.6) | 20100 (18.8) | 14071 (18.8) | 587011 (17.7) | 108977 (18.1) | 70191 (18.7) | 158014 (23.1) | 25500 (22.9) | 31539 (21) | 62152 (20.1) | 11961 (20.7) | 13287 (18.9) |
| 4 | N/A | N/A | N/A | 95383 (17.7) | 19303 (18) | 14493 (19.4) | 515136 (15.6) | 95923 (15.9) | 67639 (18) | 130251 (19) | 22699 (20.4) | 28901 (19.3) | 67827 (22) | 13663 (23.6) | 17689 (25.1) |
| 5 | N/A | N/A | N/A | 86007 (16) | 18077 (16.9) | 14529 (19.4) | 483086 (14.6) | 90963 (15.1) | 69002 (18.4) | 155241 (22.7) | 29688 (26.7) | 46151 (30.8) | 45563 (14.8) | 9678 (16.7) | 15986 (22.7) |
| Missing | 6118 (0.9) | 740 (1.0) | 363 (0.6) | 2843 (0.5) | 621 (0.6) | 487 (0.7) | 16467 (0.5) | 4478 (0.7) | 2841 (0.8) | 239 (0) | 42 (0) | 61 (0) | 24369 (7.9) | 3890 (6.7) | 4922 (7) |
| **Birth Year (calendar year)** | | | | | | | | | | | | | | | |
| 1996–2000 | 246149 (37.9) | 24183 (31.6) | 13742 (24.3) | N/A | N/A | N/A | 595764 (18.0) | 94247 (15.7) | 60493 (16.1) | N/A | N/A | N/A | 91448 (29.6) | 11497 (19.8) | 14718 (20.9) |
| 2001–2005 | 226418 (34.8) | 28696 (37.5) | 22404 (39.6) | 177464 (33) | 32891 (30.7) | 25053 (33.5) | 915788 (27.6) | 160946 (26.8) | 103809 (27.6) | 274799 (40.2) | 40639 (36.5) | 51630 (34.4) | 80982 (26.2) | 15245 (26.3) | 19903 (28.3) |
| 2006–2010 | 177388 (27.3) | 23681 (30.9) | 20421 (36.1) | 186193 (34.6) | 35865 (33.5) | 19298 (25.8) | 1355363 (40.9) | 262193 (43.6) | 158726 (42.2) | 291151 (42.6) | 49511 (44.5) | 67962 (45.3) | 95017 (30.8) | 20994 (36.2) | 25019 (35.5) |
| 2011–2015 | N/A | N/A | N/A | 174159 (32.4) | 38212 (35.7) | 30490 (40.7) | 445620 (13.5) | 84044 (14.0) | 52836 (14.1) | 117990 (17.3) | 21170 (19) | 30457 (20.3) | 41128 (13.3) | 10197 (17.6) | 10794 (15.3) |
| Missing | 0 (0) | 0 (0) | 0 (0) | 0 (0) | 0 (0) | 0 (0) | 0 (0) | 0 (0) | 0 (0) | 0 (0) | 0 (0) | 0 (0) | 0 (0) | 0 (0) | 0 (0) |
| **Season of Birth** | | | | | | | | | | | | | | | |
| Winter | 165944 (25.5) | 19262 (25.2) | 14125 (25.0) | 130149 (24.2) | 25497 (23.8) | 17480 (23.4) | 791561 (23.9) | 143631 (23.9) | 90442 (24.1) | 173455 (25.4) | 28782 (25.9) | 38507 (25.7) | 76930 (25) | 14730 (25.1) | 17393 (25.5) |
| Spring | 173843 (26.7) | 20138 (26.3) | 14850 (26.2) | 134826 (25.1) | 26508 (24.8) | 18888 (25.2) | 821434 (24.8) | 148118 (24.6) | 93382 (24.8) | 170107 (24.9) | 28040 (25.2) | 39040 (26) | 76998 (24.5) | 14458 (24.2) | 17952 (24.3) |
| Summer | 155050 (23.9) | 18893 (24.7) | 14333 (25.3) | 139463 (25.9) | 27880 (26.1) | 19654 (26.3) | 866363 (26.2) | 156746 (26.1) | 97563 (26) | 167913 (24.6) | 26640 (23.9) | 35529 (23.7) | 75452 (25.7) | 13997 (25.3) | 17121 (25.5) |
| Autumn | 155118 (23.9) | 18267 (23.9) | 13259 (23.4) | 133378 (24.8) | 27083 (25.3) | 18819 (25.1) | 833177 (25.2) | 152935 (25.4) | 94477 (25.1) | 172465 (25.2) | 27858 (25) | 36973 (24.6) | 79195 (25.7) | 14658 (25.4) | 17968 (24.7) |
| Missing | 0 (0) | 0 (0) | 0 (0) | 0 (0) | 0 (0) | 0 (0) | 0 (0) | 0 (0) | 0 (0) | 0 (0) | 0 (0) | 0 (0) | 0 (0) | 0 (0) | 0 (0) |
| **Presence of Labour** | | | | | | | | | | | | | | | |
| No | 0 (0) | 20502 (26.8) | 50049 (88.5) | N/A | N/A | N/A | N/A | N/A | N/A | N/A | N/A | N/A | 0 (0) | 8959 (15.4) | 70434 (100) |
| Yes | 649955 (100) | 56058 (73.2) | 6518 (11.5) | N/A | N/A | N/A | N/A | N/A | N/A | N/A | N/A | N/A | 308575 (100) | 48983 (84.6) | 0 (0) |

(Continued)

**Table 1.** (Continued)

| | Denmark | | | Scotland | | | England | | | New South Wales | | | Western Australia | | |
|---|---|---|---|---|---|---|---|---|---|---|---|---|---|---|---|
| Birth data: | 1997–2010 | | | 2001–2015 | | | 1 April 1998–31 March 2012 | | | 2001–2012 | | | 1996–2012 | | |
| Hospital data: | 1997–2015 | | | 2001–2016 | | | 1 April 1998–31 March 2012 | | | 2001–2012 | | | 1996–June 30 2013 | | |
| | n = 783,082 | | | n = 719,625 | | | n = 4,289,829 | | | n = 945309 | | | n = 436942 | | |
| | Vaginal Births | Emergency Caesarean Sections | Elective Caesarean Sections | Vaginal Births | Emergency Caesarean Sections | Elective Caesarean Sections | Vaginal Births | Emergency Caesarean Sections | Elective Caesarean Sections | Vaginal Births | Emergency Caesarean Sections | Elective Caesarean Sections | Vaginal Births | Emergency Caesarean Sections | Elective Caesarean Sections |
| **Apgar Score 5 Minutes** | | | | | | | | | | | | | | | |
| Median (SD) | 10 (0.5) | 10 (0.9) | 10 (0.5) | 9 (0.8) | 9 (1.1) | 9 (0.7) | N/A | N/A | N/A | 9 (0.7) | 9 (0.9) | 9 (0.8) | 9 (3.3) | 9 (1.4) | 9 (0.9) |
| **Indication for Mode of Delivery (hypertensive disorders or diabetes during pregnancy):** | | | | | | | | | | | | | | | |
| No/None coded | 621676 (95.6) | 66801 (87.3) | 51801 (91.6) | 508240 (94.5) | 93364 (87.3) | 69367 (92.7) | 2859908 (86.3) | 463617 (77.1) | 299047 (79.6) | 609667 (89.1) | 90259 (81.1) | 124554 (83) | 285303 (92.5) | 48781 (84.2) | 63072 (89.6) |
| Yes | 28279 (4.4) | 9759 (12.7) | 4766 (8.4) | 29576 (5.5) | 13604 (12.7) | 5474 (7.3) | 452627 (13.7) | 137813 (22.9) | 76817 (20.4) | 74273 (10.9) | 21061 (18.9) | 25495 (17) | 23272 (7.5) | 9152 (15.8) | 7362 (10.5) |
| **Person Years** | | | | | | | | | | | | | | | |
| Mean (SD) | 4.13 (1.6) | 3.98 (1.7) | 3.97 (1.7) | 3.79 (1.7) | 3.67 (1.7) | 3.57 (1.7) | 4.32 (1.3) | 4.27 (1.5) | 4.24 (1.6) | 4.38 (1.1) | 4.31 (1.1) | 4.27 (1.1) | 3.61 (1.8) | 3.39 (1.8) | 3.53 (1.8) |
| **Number of Infection-Related Hospitalisation Diagnoses** | | | | | | | | | | | | | | | |
| 0 | 490101 (75.4) | 55079 (71.9) | 40439 (71.5) | 430603 (80.1) | 84115 (78.6) | 58454 (78.1) | 2709444 (81.8) | 475310 (79) | 301425 (80.2) | 519026 (75.9) | 82962 (74.5) | 111119 (74.1) | 221981 (71.9) | 41293 (71.3) | 50899 (72.3) |
| 1 | 115804 (17.8) | 14756 (19.3) | 11030 (19.5) | 80925 (15) | 16645 (15.6) | 12062 (16.1) | 424269 (12.8) | 87854 (14.6) | 51187 (13.6) | 118745 (17.4) | 20115 (18.1) | 27325 (18.2) | 56106 (18.2) | 10541 (18.2) | 12744 (18.1) |
| 2 | 29238 (4.5) | 4251 (5.6) | 3217 (5.7) | 17971 (3.3) | 4080 (3.8) | 2932 (3.9) | 113368 (3.4) | 23598 (3.9) | 14451 (3.8) | 31225 (4.6) | 5432 (4.9) | 7626 (5.1) | 17549 (5.7) | 3401 (5.9) | 3973 (5.6) |
| ≥3 | 14812 (2.3) | 2474 (3.2) | 1881 (3.3) | 8317 (1.5) | 2128 (2) | 1393 (1.9) | 65454 (2) | 14668 (2.4) | 8801 (2.3) | 14944 (2.2) | 2811 (2.5) | 3979 (2.7) | 12939 (4.2) | 2698 (4.7) | 2818 (4) |

a England gestational age data for <30 weeks were not deemed reliable and excluded.

CS, cesarean; SD, standard deviation.

## Statistical analysis

Each site followed a standardised protocol for data coding and analysis to generate site-specific risk estimates for each study population, modelling risk of infection-related hospitalisation over time by mode of birth (S1 Analysis Plan). Children were followed from their birth-related hospital discharge date until an infection-related hospitalisation (up to the first 3 hospitalisations), death, emigration (where data were available), fifth birthday, or end of study period (which varied with years of available data, Table 1), whichever occurred first. Potential confounders were identified with directed acyclic graphs, which provide a visual representation of causal assumptions. Variables were defined as potential confounders if they were associated with exposure but not affected by the exposure, i.e., not an intermediate on the causal pathway, and if they were independent risk factors for the outcome. Multivariable analyses included smoking during pregnancy (no/yes), maternal age at birth (<20, 20 to <25, 25 to <30, 30 to <35, ≥35 years), parity (0, 1, 2, ≥3), gestational age (<28, 28 to <32, 32 to <34, 34 to <36, 36 to <38, 38 to 40, >40 weeks), birth weight (gestational age- and sex-specific z-score percentiles: ≤10, >10 to 25, >25 to 75, >75 to 90, >90), sex, birth year, season of birth (winter, spring, summer, and autumn), socioeconomic status (as described), and recorded hypertensive disorders or diabetes mellitus during pregnancy (no/yes). Adjusted hazard ratios (HRs) and 95% confidence intervals (CIs) were estimated using multivariate Cox proportional hazard regression models for time to first event and by Prentice, Williams, and Peterson models for recurrent events data. Recurrent events were limited to the first 3 hospitalisations as risk sets for additional hospitalisations were very small, which may result in unreliable estimates [25]. For England, smoking data were unavailable, and 30% of parity data were missing. Sensitivity analyses were conducted in the English data with and without adjusting for parity in the models and in all sites with and without adjusting for smoking. In the English data, models that did not adjust for parity were run in both the total population and in the population restricted to those with known parity status (to help disentangle any effect of parity as a confounder from any effect of missing data). For all other sites, covariates had few missing data, and no imputations were warranted.

The overall and interactive effects of labour were examined. Sites estimated infection-related hospitalisation risk for the 7 clinical infection groups. To examine how infection risk varied by child age, risks were estimated for different age periods (0 to 3, 4 to 6, 7 to 12 months, 1 to <2, 2 to <5 years of age). Finally, each site estimated infection-related hospitalisation risk in a maternal subpopulation considered at low risk for adverse outcomes, defined as cephalic-presenting infants born ≥37 weeks gestational age, with birth weight between the 10th and 90th percentiles for gestational age and sex, born to women aged 20 to 34 years without any reported hypertensive disorders or diabetes preceding or during pregnancy.

Site-specific estimates were combined in a meta-analysis. Summarised estimates included fixed and random effects models. Meta-analyses used estimates from recurrent events models, unless otherwise specified.

Sensitivity analyses assessed robustness of results. First, as we previously observed associations between prenatal antibiotic use, CS births, and childhood infection-related hospitalisation risk, we used additional variables only available in the Danish data and adjusted overall models for antibiotic use 3 months prior to and/or during pregnancy [17]. Second, to quantify unmeasured risk resulting from the lack of data on births 24 to <30 weeks gestational age in England, we restricted the Danish analysis to births ≥30 weeks gestational age. Third, as missing data were a historical issue with the English data (1998 to 2003), we compared the full English results (1998 to 2010) with restricted English results (2003 to 2012, when the issue of missing data improved). Fourth, to consider if asthma, which may be associated with CS birth

[26], was an underlying cause for the observed associations, we estimated the association between mode of birth and the separate outcomes of infection-related hospitalisation with and without a concurrent diagnosis for asthma and/or wheeze using the Western Australia data. Fifth, to examine whether the association between mode of birth and infection-related hospitalisations was similar by birth year, we estimated the overall risk stratified by birth year in 4-year intervals in the Western Australia data. Lastly, to address potential unmeasured confounding, we calculated site-specific E-values [27]. E-values estimate the minimum strength of association that an unmeasured confounder would need to have with both the exposure and outcome in order to fully explain away the observed association. Similarly, we estimated the association between mode of birth and the negative control outcome of trauma in the Western Australia data, where these hospital data were readily available and have been validated. We hypothesised that if bias or unmeasured confounding is present, we might observe an increase in risk for noninfection-related trauma admissions.

We calculated site-specific population attributable fractions to quantify the percentage and number of infection-related hospitalisations attributable to CSs.

Site-specific analyses were performed in SAS (SAS Institute, Cary, North Carolina, USA), Stata (StataCorp, College Station, Texas, USA), and R (R Core Team, Vienna, Austria), depending on the site. Meta-analyses were performed in Stata SE 16.0.

This study is reported as per the **RE**porting of studies **C**onducted using **O**bservational **R**outinely-collected **D**ata (**RECORD**) guideline (S1 Checklist).

### Ethics statement

Site-specific data use was approved by the Danish Data Protection Agency, National Health Service (NHS) Scotland Public Benefit and Privacy Panel, Central and South Bristol Multi-Centre Research Ethics Committee, New South Wales Population and Health Services Research Ethics Committee, Western Australian Department of Health Human Research Ethics Committee, Western Australian Aboriginal Health Ethics Committee, University of Western Australia Human Research Ethics Committee, and Royal Children's Hospital Human Research Ethics Committee.

## Results

In total, 7,174,787 children were identified and followed from birth-related hospital discharge until a maximum age of 5 years. Site-specific study characteristics are presented in Table 1. Overall, 1,681,966 (23%, range by jurisdiction 17% to 29%) were by CS, and of these, 727,755 (43%, range 38% to 57%) were elective. During the study period, the rates of emergency and elective CS increased in all populations. Parity, gestational age, and socioeconomic status distributions varied slightly by site (Table 1).

During follow-up, 1,502,537 children (21%) had at least 1 infection-related hospitalisation. Compared to vaginally born children, risk of infection-related hospitalisation was greater among CS-born children (HR from random effects model, HR 1.10, 95% CI 1.09 to 1.12, $p < 0.001$). The risk was higher following both elective (HR 1.13, 95% CI 1.12 to 1.13, $p < 0.001$) and emergency CS (HR 1.09, 95% CI 1.06 to 1.12, $p < 0.001$), compared with vaginal births (Fig 1). The increased risk associated with CS birth persisted through early childhood, with the highest risks for infection-related hospitalisation occurring during the first 6 months of life in children born by elective CS (Fig 2).

A higher relative risk was observed when labour did not occur compared with births following labour (HR 1.12, 95% CI 1.12 to 1.13, $p < 0.001$) (S2 Fig). When mode of birth and labour onset were jointly considered, the relative risks across exposure combinations, compared with

| Study | Total N | Cases (%) | Crude HR | | Hazard Ratio (95% CI) | % Weight (I-V) |
|-------|---------|-----------|----------|--|----------------------|-----------------|
| **Any Caesarean Section** | | | | | | |
| Denmark | 133127 | 28.3 | 1.17 | | 1.12 (1.11, 1.13) | 12.57 |
| Scotland | 164785 | 21.8 | 1.11 | | 1.09 (1.08, 1.11) | 8.70 |
| England | 946280 | 21.1 | 1.1 | | 1.08 (1.08, 1.09) | 51.38 |
| New South Wales | 299560 | 26.7 | 1.13 | | 1.11 (1.10, 1.12) | 20.03 |
| Western Australia | 128367 | 24.3 | 1.02 | | 1.12 (1.10, 1.13) | 7.33 |
| I-V Subtotal  (I-squared = 92.5%, p = 0.000) | | | | | 1.10 (1.09, 1.10) | 100.00 |
| D+L Subtotal | | | | | 1.10 (1.09, 1.12) | |
| | | | | | | |
| **Emergency Caesarean Section** | | | | | | |
| Denmark | 76560 | 28.1 | 1.17 | | 1.12 (1.10, 1.13) | 13.67 |
| Scotland | 96556 | 21.5 | 1.11 | | 1.08 (1.06, 1.10) | 9.85 |
| England | 577274 | 20.9 | 1.09 | | 1.05 (1.04, 1.06) | 51.99 |
| New South Wales | 127756 | 26.3 | 1.11 | | 1.09 (1.07, 1.10) | 17.39 |
| Western Australia | 57933 | 24.8 | 1.1 | | 1.11 (1.09, 1.13) | 7.09 |
| I-V Subtotal  (I-squared = 95.6%, p = 0.000) | | | | | 1.07 (1.07, 1.08) | 100.00 |
| D+L Subtotal | | | | | 1.09 (1.06, 1.12) | |
| | | | | | | |
| **Elective Caesarean Section** | | | | | | |
| Denmark | 56567 | 28.5 | 1.18 | | 1.13 (1.11, 1.14) | 13.18 |
| Scotland | 68229 | 22.2 | 1.12 | | 1.11 (1.09, 1.13) | 8.64 |
| England | 369006 | 21.4 | 1.11 | | 1.13 (1.13, 1.14) | 42.17 |
| New South Wales | 171804 | 26.9 | 1.15 | | 1.12 (1.11, 1.13) | 27.53 |
| Western Australia | 70434 | 23.9 | 1.03 | | 1.13 (1.11, 1.15) | 8.48 |
| I-V Subtotal  (I-squared = 35.6%, p = 0.184) | | | | | 1.13 (1.12, 1.13) | 100.00 |
| D+L Subtotal | | | | | 1.13 (1.12, 1.13) | |

**Fig 1. Site-specific and meta-analysis HRs for infection-related hospitalisations.** Estimates are from recurrent events models fitted for total time. Models adjusted for sex, gestational age, birth weight z-score, smoking during pregnancy, maternal age at birth, parity, area level deprivation, birth year, medical indication for type of delivery, and season of birth. Reference is vaginal births. CI, confidence interval; D+L, DerSimonian and Laird random effects model; HR, hazard ratio; I-V, inverse-variance weighted fixed effects model.

vaginal births, were similar in size to the overall findings (S2 Table). In the low-risk maternal subpopulation, the overall relative risk of infection-related hospitalisation in children born by CS was similar to overall findings (elective CS HR 1.14, 95% CI 1.12 to 1.15, $p < 0.001$; emergency CS HR 1.08, 95% CI 1.04 to 1.12, $p < 0.001$) (Fig 3).

Increased risks for hospitalisation were observed in all clinical infection groups. For upper and lower respiratory tract, viral, gastrointestinal infections, and genitourinary infections, the highest risk estimates were for elective CS (Fig 4).

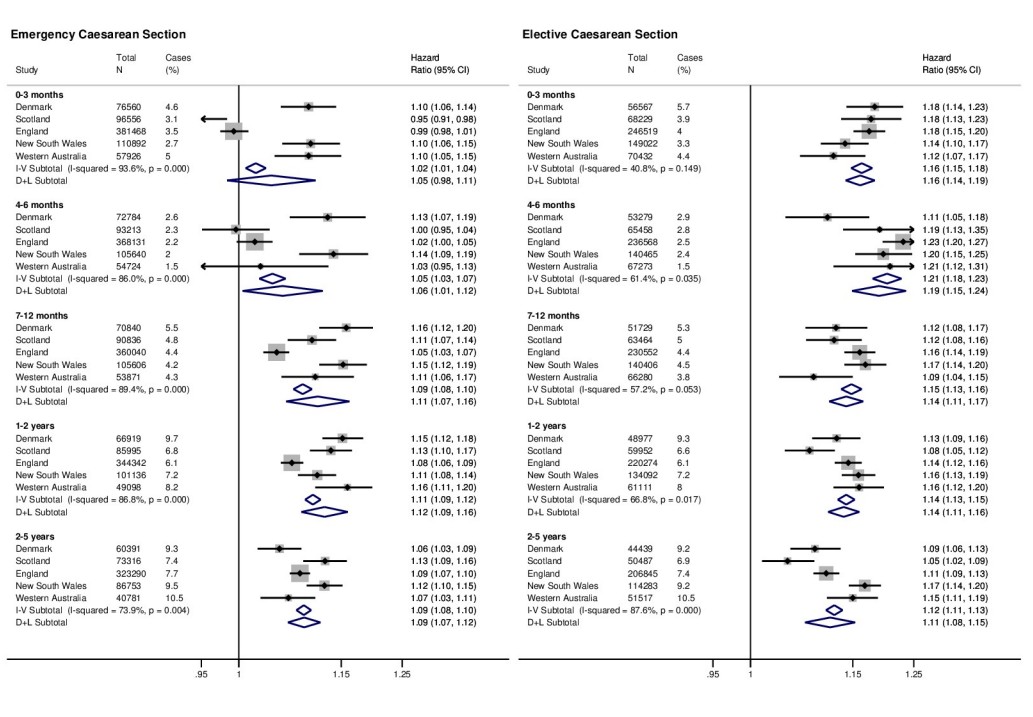

**Fig 2. Site-specific and meta-analysis HRs for infection-related hospitalisations by age at first occurrence.** Estimates are from first event models adjusted for sex, gestational age, birth weight z-score, smoking during pregnancy, maternal age at birth, parity, area level deprivation, birth year, medical indication for type of delivery, and season of birth. Reference is vaginal births. Estimates for England based on follow-up time until first infection-related hospitalisation admission as data on exact date of birth were not available. CI, confidence interval; D+L, DerSimonian and Laird random effects model; HR, hazard ratio; I-V, inverse-variance weighted fixed effects model.

No substantial differences were observed in sensitivity analyses (S3 and S4 Tables, S3 Fig). Analyses restricted to infection-related hospitalisations either with or without a concurrent asthma and/or wheeze diagnosis were similar to the overall analyses that included all infection-related hospitalisations (S5 Table). Risks for infection-related hospitalisation stratified by birth year were similar with overlapping CIs (S6 Table). Based on the E-values, an unmeasured confounder would need a moderate association of at least 1.20 (depending on the study site and type of CS) with both mode of birth and infection-related hospitalisation in the child, in order to explain away the observed associations (S7 Table). As a reference point from the Danish data, maternal age has an association of 1.27 with mode of birth, yet the association with infection in the child is only 0.90. Similarly, the association of smoking during pregnancy with mode of birth is 0.97, but with infection in the child is 1.28. Hypertensive disorders and diabetes during pregnancy is strongly associated with mode of birth (2.69) and moderately associated with infection in the child (1.19). In the analysis of mode of birth and the negative control outcome of trauma admissions, we did not observe an association (S8 Table).

Population attributable fractions are provided in S9 Table. If the associations are causal, 2% to 3% of children with infection-related hospitalisations could attribute their infection to being born by CS. Among the 1,502,537 children who had an infection-related hospitalisation in our study, about 14,000 calculated children had infections that could be attributed to being born by emergency CS and 18,500 to being born by elective CS.

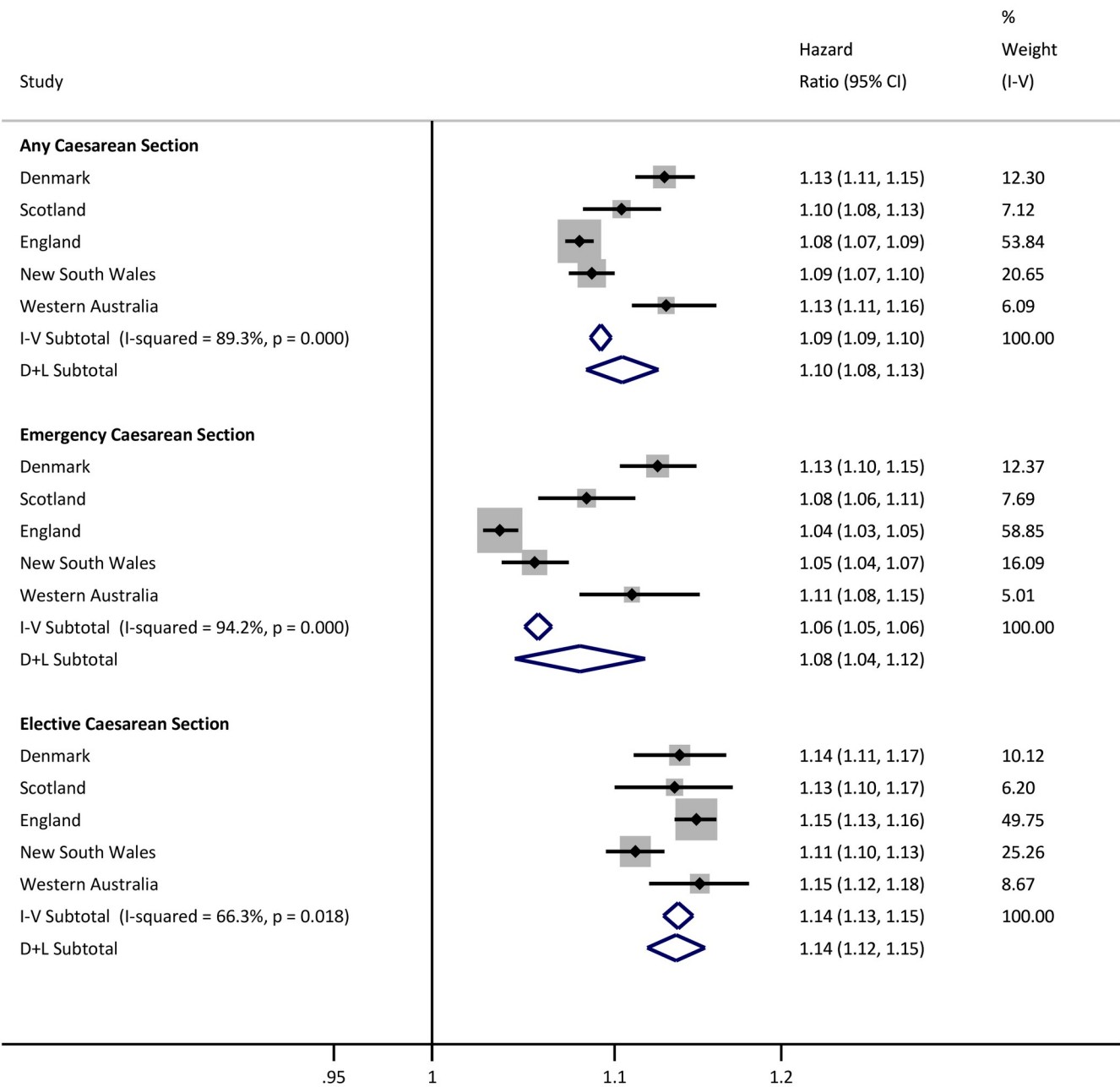

**Fig 3. Site-specific and meta-analysis HRs for infection-related hospitalisations in children born to a low-risk population of mothers.** Estimates are from recurrent events models fitted for total time. Models adjusted for sex, gestational age, birth weight z-score, smoking during pregnancy, maternal age at birth, parity, area level deprivation, birth year, medical indication for type of delivery, and season of birth. Reference is vaginal births. Low-risk population of births defined as cephalic presenting infants born ≥37 weeks gestational age, with birth weight between the 10th and 90th percentiles for gestational age and sex, born to women aged 20 to 34 years without any reported medical conditions during pregnancy. CI, confidence interval; D+L, DerSimonian and Laird random effects model; HR, hazard ratio; I-V, inverse-variance weighted fixed effects model.

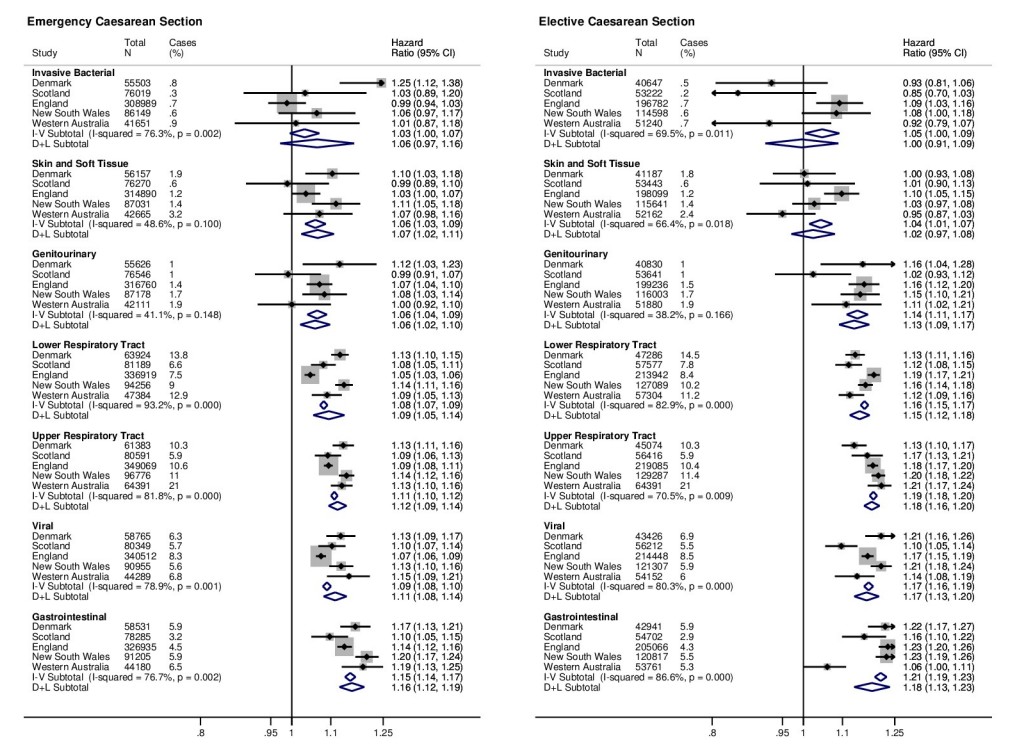

**Fig 4. Site-specific and meta-analysis HRs for infection-related hospitalisations by clinical infection group.** Estimates are from recurrent events models fitted for total time. Models adjusted for sex, gestational age, birth weight z-score, smoking during pregnancy, maternal age at birth, parity, area level deprivation, birth year, medical indication for type of delivery, and season of birth. Reference is vaginal births. CI, confidence interval; D+L, DerSimonian and Laird random effects model; HR, hazard ratio; I-V, inverse-variance weighted fixed effects model.

## Discussion

In this multinational, population-based study, CS was associated with an approximately 10% increased risk of infection-related hospitalisation in offspring up to 5 years of age compared with vaginal birth. Across the 5 populations, risk estimates were comparable, and both elective and emergency CS were associated with infection-related hospitalisation, with the highest increased risk (13%) following elective CS. The findings point to causal determinants of susceptibility to infection shared by CS-born children, regardless of underlying indications.

The current study population is drawn from comparable high-income settings where paediatric hospital care is free, but which vary considerably with respect to obstetric practice, CS rates, proportion of emergency and elective CS, and the use of public and private obstetric care. We used a standardised protocol to eliminate methodological differences and the use of total population data reduces selection bias. In regression models, the results were not explained by known risk factors for infection, including maternal smoking, socioeconomic status, parity, birth weight, gestational age, or season of birth, and this is corroborated by similar results across populations where the distribution of these factors vary.

Our findings are in keeping with smaller population-level, single-jurisdiction studies that largely focus on specific infection outcomes, such as lower respiratory tract and gastrointestinal infections and which show that CS is associated with increased infection-related hospitalisation, and (where data are available) with greater risk following elective CS [11–15]. In contrast to previous studies, we attempted to address potential confounding by a series of

sensitivity analyses. We analysed a predefined subpopulation considered at low obstetric risk in whom findings were similar to the overall cohort, indicating that confounding by indication was unlikely to be responsible for the observed associations. In addition, where data were available, we analysed a negative control outcome (hospitalisation with trauma) and observed no association with mode of birth, which gives confidence that our findings are not reflective of bias or unmeasured confounding. We adjusted for concomitant asthma and/or wheeze, which has been repeatedly associated with CS birth [26], and which may contribute to the observed associations between birth mode and infections; findings were largely unchanged. Finally, in the Danish population, where pregnancy antibiotic data were available, we adjusted for antibiotic exposure prior to and/or during pregnancy and the findings were unchanged (S4 Table).

Our study has a number of additional strengths. We restricted infection-related hospitalisations to readmission following the birth-related discharge to avoid bias from suspected neonatal sepsis that may result from direct microbial exposure during birth. The outcome was infection-related hospitalisation, which minimises differences observed in primary care or emergency department presentations that may reflect social gradients in health literacy, health-seeking behaviour, or physician management rather than clinical severity [28]. We acknowledge important limitations. Models were adjusted for confounders and we performed extensive sensitivity analyses, but residual, unmeasured confounding is possible. However, as illustrated in Fig 1, there was not a consistent pattern across the 4 countries that associations weaken upon adjustment. Second, there have been changes in clinical practice and diagnostic coding, demographics, and lifestyle that cannot be fully measured or accounted for over the study period. These include changes in obstetric guidelines, use of antenatal steroids in threatened preterm delivery and elective CS, timing of antibiotic prophylaxis relative to cord clamping at CS, and differences in covariate definitions, such as socioeconomic status. Guidelines for the use of prophylactic antibiotics at CS changed during the study period, but we were unable to quantify any impact. In Denmark, United Kingdom, and Australia, perioperative broad-spectrum antibiotics are recommended for CS, and after 2010, guidelines changed from administration after cord clamping to preincision. The UK National Institute for Health and Care Excellence (NICE) guidelines changed in 2011 [29], Danish guidelines in 2012 [30], and current Australian guidelines [31] support use of preincision antibiotics but are less prescriptive. Peripartum antibiotics may have relatively subtle microbial effects for the infant [32]. In our study, 83% of CS births occurred before 2011 and offspring would not have been exposed to antibiotics at delivery. Third, availability and definitions of some data varied between centres. We addressed these differences with sensitivity analyses where possible, and results were essentially unchanged. Categorisation of emergency and elective CSs was based on available data in the birth databases. Timing of CS relative to onset of labour was not available for England, Scotland, or New South Wales. However, Scottish national coding rules state that scheduled elective CSs that occur during labour should be recorded as emergency CSs. Fourth, although the approximate proportion of births in public versus private facilities is known for each country (e.g., 26% of births are in private hospitals in Australia [33] compared to virtually none elsewhere), individual-level facility data were unavailable, and the relationship with mode of birth is unknown. Finally, data on infections managed in primary care or in emergency departments and on postnatal factors that influence infection risk, such as infant feeding, vaccination status, and postnatal smoke exposure, were unavailable. If these varied by mode of birth, then this may affect the observed associations.

The current study did not address the mechanisms underlying the epidemiological observations, but our findings inform future research directions. Further studies in other settings, particularly low- and middle-income countries, are needed and should include data on modifiable postnatal exposures. Mechanistic studies may guide the development of

interventions. Differences in initial microbial exposure by mode of birth, which may persist for months or possibly years [34], may contribute to the increased risk of infection-related hospitalisation following CS by effects on the development of postnatal immune responses. The composition and function of the early microbiome have been linked to a range of adverse short- and longer-term immune-mediated outcomes [9], although we are not aware of studies that have directly linked the postnatal microbiome with risk of common childhood infections. Infection-related hospitalisation was increased after elective CS, when membranes are usually intact at delivery, and for infections of sites where direct inoculation of maternal microbiome during vaginal delivery may be important in early protective mucosal immunity in the gastrointestinal and respiratory tracts [10,35]. Additional explanations include possible effects of short-term antenatal corticosteroids given to mothers delivering via CS to reduce infant respiratory morbidity, which was recommended practice in all jurisdictions (apart from Denmark) during the latter part of the study period. Corticosteroids are broadly immunosuppressive, but there is little evidence that antenatal steroids affect the incidence of postnatal infection and individual-level data on corticosteroid exposure were unavailable. Finally, unmeasured heritable and shared environmental factors may contribute to the observed associations.

Our findings have implications for clinical practice and public health policy. CS rates are increasing and exceed international recommendations [1]. In 2010, WHO estimated the cost of "excess" CSs to be approximately US$2.32 billion [36]. Infection is the leading cause of early childhood hospitalisation, and this potential risk should be considered when discussing obstetric management, especially if vaginal birth is clinically safe and appropriate. In addition, healthcare costs of potentially avoidable childhood infection-related hospitalisation are likely to be considerable.

In conclusion, in a large multinational study, children born by CS were at increased risk of infection-related hospitalisation until age 5 years. These findings may contribute to the global effort to reduce the rates of CS that are not medically indicated.

## Supporting information

**S1 Checklist. RECORD checklist.**
(DOCX)

**S1 Analysis Plan. Analysis plan for study populations.**
(DOCX)

**S1 Fig. Flowcharts for site-specific study populations.**
(DOCX)

**S2 Fig. Site-specific and meta-analysis hazard ratios for infection-related hospitalisation in births without labour.** Estimates are from recurrent events models fitted for total time. Models adjusted for sex, gestational age, birth weight z-score, smoking during pregnancy, maternal age at birth, parity, area level deprivation, birth year, medical indication for type of delivery, and season of birth. Reference is births with labour. D+L, DerSimonian and Laird random effects model; I-V, inverse-variance weighted fixed effects model.
(DOCX)

**S3 Fig. Site-specific and meta-analysis hazard ratios for infection-related hospitalisation, not adjusted for smoking during pregnancy.** Estimates are from recurrent events models fitted for total time. Models adjusted for sex, gestational age, birth weight z-score, maternal age at birth, parity, area level deprivation, birth year, medical indication for type of delivery, and season of birth. Reference is vaginal births. D+L, DerSimonian and Laird random effects

model; I-V, inverse-variance weighted fixed effects model.
(DOCX)

**S1 Table. Variable definition.**
(DOCX)

**S2 Table. Risk of infection-related hospitalisation by mode of birth and presence of labour.** Estimates not available for New South Wales (Australia), Scotland, and England. Estimates are from recurrent events models fitted for total time. Models adjusted for sex, gestational age, birth weight z-score, smoking during pregnancy, maternal age at birth, parity, area level deprivation, birth year, medical indication for type of delivery, and season of birth.
(DOCX)

**S3 Table. Sensitivity analysis—Parity adjustments and restricted birth years, English data.** Estimates are from recurrent events models fitted for total time. Fully adjusted model adjusts for sex, gestational age, birth weight z-score, maternal age at birth, parity (except for models that specify that parity was not adjusted for), area level deprivation, birth year, medical indication for type of delivery, and season of birth.
(DOCX)

**S4 Table. Sensitivity analysis—Prenatal antibiotic use and gestational age, Danish data.** Estimates are from first event models adjusted for: sex, gestational age, birth weight z-score, smoking during pregnancy, maternal age at birth, parity, area level deprivation, birth year, medical indication for type of delivery, and season of birth.
(DOCX)

**S5 Table. Sensitivity analysis—Hazard ratios for infection-related hospitalisations with and without a concurrent diagnosis of asthma and/or wheeze, Western Australia data.** [*]All other infection-related hospitalisation excluded from the analyses. Asthma was identified with ICD-10 code J45; wheeze was identified with ICD-10 code R06.2. Estimates are from recurrent events models fitted for total time. Models adjusted for: sex, gestational age, birth weight z-score, smoking during pregnancy, maternal age at birth, parity, area level deprivation, birth year, medical indication for type of delivery, and season of birth.
(DOCX)

**S6 Table. Sensitivity analysis—Risk of infection-related hospitalisation by birth year, Western Australia data.** Estimates are from first event models adjusted for: sex, gestational age, birth weight z-score, smoking during pregnancy, maternal age at birth, parity, area level deprivation, birth year (overall estimate only), medical indication for type of delivery, and season of birth.
(DOCX)

**S7 Table. Sensitivity analysis—E-values.** Calculated E-values for hazard ratios with outcome prevalence >15%. The E-value is defined as the minimum strength of association, on the risk ratio scale, that an unmeasured confounder would need to have with both the treatment and the outcome to fully explain away a specific treatment–outcome association, conditional on the measured covariates. VanderWeele TJ, Ding P. Sensitivity Analysis in Observational Research: Introducing the E-Value. Ann Intern Med. 2017;167(4):268–74.
(DOCX)

**S8 Table. Sensitivity analysis—Hazard ratios for trauma hospitalisations, Western Australia data.** Estimates are from first event models adjusted for: sex, gestational age, birth weight z-score, smoking during pregnancy (unless specified), maternal age at birth, parity, area level

deprivation, birth year, medical indication for type of delivery, and season of birth.
(DOCX)

**S9 Table. Population attributable fractions.** Population attributable fractions are defined as the proportion of all cases (i.e., children admitted to hospital with an infection) in the population that could be attributed to the exposure (i.e., cesarean section). *Subpopulation of emergency cesarean section and vaginal births and cases only. Elective cesarean section births and cases are excluded. **Subpopulation of elective cesarean section and vaginal births and cases only. Emergency cesarean section births and cases are excluded. Mansournia Mohammad Ali, Altman Douglas G. Population attributable fraction. BMJ. 2018;360:k757.
(DOCX)

## Acknowledgments

The authors would like to thank Professor Fiona Stanley AC for her guidance and support.

## Author Contributions

**Conceptualization:** Jessica E. Miller, Raphael Goldacre, Hannah C. Moore, Justin Zeltzer, Marian Knight, Carole Morris, Sian Nowell, Rachael Wood, Kim W. Carter, Parveen Fathima, Nicholas de Klerk, Tobias Strunk, Jiong Li, Natasha Nassar, Lars H. Pedersen, David P. Burgner.

**Data curation:** Jessica E. Miller.

**Formal analysis:** Jessica E. Miller, Raphael Goldacre, Justin Zeltzer, Sian Nowell, Parveen Fathima.

**Funding acquisition:** Jessica E. Miller.

**Investigation:** Jessica E. Miller.

**Methodology:** Jessica E. Miller, Marian Knight, Nicholas de Klerk, Lars H. Pedersen, David P. Burgner.

**Project administration:** Jessica E. Miller.

**Resources:** Hannah C. Moore, Marian Knight, Carole Morris, Jiong Li, Natasha Nassar.

**Supervision:** Jessica E. Miller, David P. Burgner.

**Validation:** Jessica E. Miller.

**Writing – original draft:** Jessica E. Miller, Marian Knight, Lars H. Pedersen, David P. Burgner.

**Writing – review & editing:** Jessica E. Miller, Raphael Goldacre, Hannah C. Moore, Justin Zeltzer, Carole Morris, Sian Nowell, Rachael Wood, Kim W. Carter, Parveen Fathima, Nicholas de Klerk, Tobias Strunk, Jiong Li, Natasha Nassar.

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
