## [Editor Report · Decision Letter 0]

10 Jan 2020

Dear Dr Miller, 

Thank you for submitting your manuscript entitled "Mode Of Birth And Risk Of Infection-Related Hospitalisation In Childhood: A Total Population Study Of 7.17 Million Births From Four Countries" for consideration by PLOS Medicine.

Your manuscript has now been evaluated by the PLOS Medicine editorial staff and I am writing to let you know that we would like to send your submission out for external peer review.

Please re-submit your manuscript within two working days, i.e. by 14th Jan 2020, 11:59PM.

Kind regards,

Louise Gaynor-Brook, MBBS PhD

Associate Editor

PLOS Medicine

---

## [Decision Letter · Decision Letter 1]

16 Jun 2020

Dear Dr. Miller,

Thank you very much for submitting your manuscript "Mode Of Birth And Risk Of Infection-Related Hospitalisation In Childhood: A Total Population Study Of 7.17 Million Births From Four Countries" (PMEDICINE-D-20-00075R1) for consideration at PLOS Medicine. 

Your paper was evaluated by a senior editor and discussed among all the editors here. It was also discussed with an academic editor with relevant expertise, and sent to four independent reviewers, including a statistical reviewer. The reviews are appended at the bottom of this email and any accompanying reviewer attachments can be seen via the link below:

[LINK]

In light of these reviews, I am afraid that we will not be able to accept the manuscript for publication in the journal in its current form, but we would like to consider a revised version that addresses the reviewers' and editors' comments. Obviously we cannot make any decision about publication until we have seen the revised manuscript and your response, and we plan to seek re-review by one or more of the reviewers. 

We expect to receive your revised manuscript by Jul 07 2020 11:59PM. Please email us (plosmedicine@plos.org) if you have any questions or concerns.

We look forward to receiving your revised manuscript. 

Sincerely,

Thomas McBride, PhD

Senior Editor 

PLOS Medicine

plosmedicine.org

1- Please adjust the Title slightly: “Mode Of Birth And Risk Of Infection-Related Hospitalisation In Childhood: A Population Cohort Study Of 7.17 Million Births From Four Countries”

2- Thank you for acknowledging the limits on sharing the data used in this study. Please update your data statement to include contact information (email or website) where readers may request access to each of the datasets used here.

3- Please ensure that the study is reported according to the STROBE or RECORD guideline, and include the completed checklist as Supporting Information. 

Please add the following statement, or similar, to the Methods: "This study is reported as per the Strengthening the Reporting of Observational Studies in Epidemiology (STROBE) guideline (S_ Checklist)."

The RECORD guideline can be found here: https://www.equator-network.org/reporting-guidelines/record/

4- Did your study have a prospective protocol or analysis plan? Please state this (either way) early in the Methods section.

5- Please structure your abstract using the PLOS Medicine headings (Background, Methods and Findings, Conclusions).

6- In the abstract and throughout the manuscript, is it more accurate to describe the cohort as “all *recorded* singleton live births” or similar?

7- In the Abstract Methods and Findings section, please provide the full date range (Jan 1 1996– Dec 31 2015?), and clarify that not all cohorts covered this entire range.

8- In the Abstract and throughout the manuscript, please include p-values alongside the 95% CIs.

9- In the last sentence of the Abstract Methods and Findings section, please describe the main limitation(s) of the study's methodology.

10- In the Abstract Conclusions, please address the study implications without overreaching what can be concluded from the data; the phrase "In this study, we observed ..." may be useful.

11- At this stage, we ask that you include a short, non-technical Author Summary of your research to make findings accessible to a wide audience that includes both scientists and non-scientists. The Author Summary should immediately follow the Abstract in your revised manuscript. This text is subject to editorial change and should be distinct from the scientific abstract. Please see our author guidelines for more information: https://journals.plos.org/plosmedicine/s/revising-your-manuscript#loc-author-summary

12- In the Results section, please include the unadjusted HRs along with the adjusted HRs.

13- Please re-organize the Discussion and present as follows: a short, clear summary of the article's findings; what the study adds to existing research and where and why the results may differ from previous research; strengths and limitations of the study; implications and next steps for research, clinical practice, and/or public policy; one-paragraph conclusion.

14- In the Conclusion, please be a bit more specific regarding the implications and recommendations for future research.

15- Please place reference numbers in square brackets preceding punctuation. Please also make sure to use the "Vancouver" style for reference formatting, and see our website for other reference guidelines https://journals.plos.org/plosmedicine/s/submission-guidelines#loc-references

16- Please provide titles and legends for each individual table and figure in the Supporting Information and list all SI files at the end of the manuscript text.

Comments from the reviewers:

Reviewer #1: Miller et al. present the findings of a multi-country analysis evaluating the association between children born by CS compared to vagina delivery and infection-related hospitalisation. The statistical methods are straightforward, using multivariate Cox regression models to determine the AHRs for harmonised across country-specific data, and pooled into an aggregate figure in a two-stage process used fixed and random-effects meta-analysis. The results are largely consistent with previous studies, but this study does have a much larger sample size, include broader range of IRHs, and use some additional methods to address residual confounding, including using DAGs to graphically identify confounders, and E-values to measure the potential effects of unmeasured confounding on the results. The results are quite consistent across all four countries, and should generalise to developed countries with similar quality of obstetrics care. 

Major comments

1) I thought what would strengthen the case for this paper is the authors could elaborate a bit further in the discussion what this particular adds compared to previous Australian (ref 13) and Danish studies (ref 7). Both these previous studies are not small either (Australian study n = 212k; Danish study n = 750k) which also quite similar results (11-20% increased risk). That means highlighting what the limitations of these previous studies may have been and what this current study has improved upon. As it stands, the study feels largely like a confirmatory study of previous results (which itself is important aspect of science) but I think the authors could enhance the novelty of their approach a bit further. 

2) The primary limitation the authors eluded to in the introduction and discussion (which I agree with) is confounding by indication. As a randomised trial is clearly not possible, a causal relationship remains elusive. Whilst the authors have given this thought, through use of DAGS to identify potential confounders and quantify the influence through E-values, the main multivariate statistical analysis does not actually attempt to address this issue. I thought there was a clear opportunity here to use a propensity score adjustment in the model (https://www.ahajournals.org/doi/10.1161/CIRCOUTCOMES.113.000359) either using a matching process or just direct adjusting for the PS. The extraordinary size of the sample also creates an opportunity for covariate balancing (https://rss.onlinelibrary.wiley.com/doi/full/10.1111/rssb.12027). 

This is not to say what the authors did in the analysis was incorrect (as their approach is entirely valid) but rather that their primary limitation could be partially addressed by consideration of the above statistical approaches. 

3) I would have liked to seen some thought given to using a negative control outcome (https://www.ncbi.nlm.nih.gov/pmc/articles/PMC5428075/) to further strengthen the causal inference framework. Non-infection related hospitalisations could potentially be used here (the inverse of the outcome). The hypothesized mechanism would be that any sources of unmeasured confounding would also lead to an increase in non-infection related hospitalisation. An expected null results in the negative control outcome would remarkably strengthen the results. 

Minor:

4) Abstract methods: specify whether pooled results following in the findings section of the abstract are from fixed or random-effects meta-analysis

5) Methods - statistical analysis: Use of DAGS - I agree that DAGS are useful tools here for causal inference as a graphic tool but I haven't seen where the authors specifically elaborated how they were used to identify confounder in the multivariate analysis. Please elaborate on variable selection and criteria applied. 

6) Methods - sensitivity analysis: Was the sensitivity analysis on English data conducted with imputation of parity or was this as complete case analyses? And imputation was used - please specify the exact methodology. 

Reviewer #2: The authors analyze databases and show a consistently higher incidence of hospitalization due to infection up to 5 years of age in children born by CS versus those born by vaginal delivery in four wealthy countries. 

The manuscript is clear, concise and well written.

Major comments

1. I did not find the argument convincing that changes in gut flora are likely the explanation for the differences in infection rates. My objections are:

a) Do we really know that differences in gut flora following C/S persist for years?

b) What is the direct evidence that gut flora plays any role in what infections a child gets?

c) No other theories are presented. Is it possible that women who end up with a C/S seek health care for their children more often than do women who had a vaginal delivery? The authors imply that hospitalization is an objective outcome. However there are women who do not want any intervention for themselves or for their children so are less likely to have a C/S or to have their children admitted with infection. 

2. I would make it a bit clearer that although the incidence of infection increased 10% with CS, only approximately 2% of children had an excess admission related to being born by CS. It might be helpful to tell the reader how many excess admissions this would be annually for each of the 5 jurisdictions that were studied. 

Minor comments

3. I would clarify in the abstract that infections occurring during the birth hospitalization were excluded. 

4. "An increased risk of specific infection-related hospitalisations, mainly lower respiratory tract and gastrointestinal infections, has been associated with CS.(11, 12)" Please provide details from the quoted studies.

5. Why were children with congenital malformations excluded? With this sample size, findings should not be affected by inclusion of children who might have a higher incidence of infections. I am not suggesting that the authors re-do the study but they ought to explain to the reader why they did this. 

6. Why did the authors only look at the first 3 infection related hospitalizations?

7. When did data collection end? How many children were followed for less than 5 years simply because data collection ended?

8. The terms "birth following labour" and "emergency CS" are both used. Assuming that they refer to the same thing, consistent terminology should be used throughout the manuscript. 

Reviewer #3: Thank you for allowing me to review this manuscript.

Here, Miller and colleagues present the results from a large multicenter registry based study on the associations between CS delivery and childhood risk of hospitalizations with infectious cause in the first five years of life. The topic of research is not in itself novel, but never before has this association been shown in such a large cohort. I must congratulate the authors in collecting this huge number of women/children for analysis. The text is clearly written and the study is well performed, though some concerns do arise. The sub-analyses are also interesting to evaluate potential mechanisms and diverging associations. 

CS delivery has time and time again been associated with asthma development in childhood. Asthma is almost always preceded by asthmatic episodes typically initiating in the first years of life and the number one cause of hospitalization in children. Asthmatic episodes are almost always triggered by viral or bacterial infections in this window. Could asthma be the underlying cause for the associations observed? I would like to see the associations adjusted for childhood asthma and stratified for hospitalizations with/without concurrent asthma/wheeze.

Children born by CS are more often hospitalized right after birth and also more often treated for infections here. Was this potential hospital stay associated with the outcomes? E.g. the CS might only associate with later infections if the child were hospitalized after birth because of infections or other complications. 

Microbiome hypothesis may be appealing, but other than differences between elective and emergency CS you have not presented much data to support this, so it comes off as speculative. To get closer to whether the associations might actually be caused by microbial derangements after CS, it would be a great strength to show that similar outcome associations existed among vaginal delivered children whose mothers were treated during birth.

A potential issue, when evaluating associations in such large datasets are the potential of insignificant significance. Would it be possible to calculate a population attributable risk fraction (PARF)? 

Maybe I missed it, but how did antibiotics to the mother in pregnancy affect the associations?

No line numbers in manuscript make specific comments less trivial...

"gestational age (<28, 28-<32, 32-<34, 34-<36, 36-<38, 38-40, >40 weeks)", I would suggest more detailed i.e. weekly categorization around 37-40.

A major difference between elective and emergency CS is that the birth induction is natural in most emergency sections, whereas in many countries elective is performed at 38+0. Gestational age is a major factor for perinatal outcomes - the two week difference may carry a large part of the effect.

Since you are doing Cox regression I suggest you try various stratified Cox regression. a) stratified by year of birth; b) stratified by gestational age in detailed categories. 

How do you account for increasing rates of CS if you do not adjust for birth year? I suggest you stratify by birth year.

The association between prenatal antibiotics and off-spring outcomes may be confounded (check out PMID: 25066330 also in Danish registry data showing the effect of prenatal antibiotics is not likely to be causal).

"When mode of birth and labour onset were jointly considered, the relative risks across exposure combinations, compared with vaginal births, were similar in size to the overall findings (Table S2)." → I do not agree. Your table S2 shows a similar difference in risk between pre/post labour as between elective/emergency CS in the Danish data.

In the Danish data you should be able to distinguish between induced and spontaneous delivery similar to the WA and not just from the CS coding. I don't believe your table S2 of DK vs WA are showing comparable data.

How is labour defined in DK in figure S2?

It seems to be just as large an effect of no-labour as of CS -- this does not lend support to your microbiome interpretation.

Figure 3 is not well explained -- who is the low risk group? The reference group?

In the discussion you mention the changes in policy - why not investigate the effects of these? That could be very interesting and straight-forward in your data?

Generally the whole discussion seems a bit too long and not on point with your data. Too much unfounded mechanistic talk. I would suggest more testing and discussion on the differences between countries? Could some of it be explained or elicited? As such it seems like your overall difference between elective/emergency CS is carried by England. This should be discussed in detail. And on the same line of thoughts, you see very similar estimates in DK: why?

Missing labour in table 1.

Small detail but some of the references background study are based on the same registry data that you are examining (ref 7).

"population-specific obstetric practice may have unknown effects" -- could you not investigate this?

Reviewer #4: May 2020

Review- mode of birth and risk of infection releated hospitalization in childhood

Thank you for the opportunity to review this manuscript.

This study explores the prevalence of infection releated hospitalization in childhood according to delivery mode. Including over 7 million patients from four different countries. The HR was higher in the CS group for all infection related hospitalizations. 

General remarks:

This is a overall interesting study with a large study size and a well thought out study protocol, there a re only a few remarks:

1. In the limitation section, please add that there is only record of hospitalizations and there is no data regarding infections which were treated ambulatory.

2. In the discussion you mention corticosteriod treatment as a part of the treatment protocol in all elective CS, is this true? If so, I would add that it is not a long term treatment.

3. Sheiner et all published a very similar study last year, this study too had a large sample size, please refer to this study with similarities and differentces. By the way, in his study population, corticosteriod population is not given automatically in all cases of elective CS.

[LINK]

---

## [Decision Letter · Decision Letter 2]

29 Sep 2020

Dear Dr. Miller,

Thank you very much for re-submitting your manuscript "Mode Of Birth And Risk Of Infection-Related Hospitalisation In Childhood: A Population Cohort Study Of 7.17 Million Births From Four Countries" (PMEDICINE-D-20-00075R2) for review by PLOS Medicine.

I have discussed the paper with my colleagues and the academic editor and it was also seen again by three reviewers. I am pleased to say that provided the remaining editorial and production issues are dealt with we are planning to accept the paper for publication in the journal.

[LINK]

We look forward to receiving the revised manuscript by Oct 06 2020 11:59PM. 

Sincerely,

Thomas McBride, PhD

Senior Editor 

PLOS Medicine

plosmedicine.org

Comments from the Academic Editor:

Editor request #12 was to add unadjusted HRs. But they have declined to do this although they do present them in Figure 1 and the pattern is fairly reassuring. I think this is worthy of comment in the discussion. The big issue with weak associations like these is they are much more likely to be due to unmeasured confounders. One of the markers for unmeasured confounders is whether adjustment for known confounders had an effect. i.e. you are more likely to believe that an adjusted HR of 1.10 might be causal if the unadjusted HR = 1.10 than if it was 1.40. 

Figure 1 shows that there is not a consistent pattern across the 4 countries that associations weaken on adjustment. I think that they should flag the analysis of unadjusted HRs in the part of the discussion where they mention unmeasured confounders as they could easily be missed otherwise.

1- Thank you for updating your data statement. However, PLOS policy does not allow authors to be the point of contact for data requests. Please replace Lars Pedersen with a different (non-author) contact, such as an institutional data officer, or a member of the IRB.

2- Title, please add “high-income”: "Mode Of Birth And Risk Of Infection-Related Hospitalisation In Childhood: A Population Cohort Study Of 7.17 Million Births From Four *High-income* Countries"

3- Please remove the “Role of the funding source” section from the main text.

4- Please remove italics from the References.

5- Unless there was a statistical test comparing the risk of infection between elective and emergency CS, please do not report this as a difference. The Abstract lines 67-71 would read better as:

“Compared to vaginally-born children, risk of infection was greater among CS-born children (hazard ratio from random effects model, HR 1.10, 95% CI 1.09-1.12, p< 0.001). The risk was higher following both elective (HR 1.13, 95% CI 1.12-1.13, p< 0.001) and emergency CS (HR 1.09, 95% CI 1.06-1.12, p<0.001).” Please also rephrase for the same data presented in the results, Lines 305-209.

6- Table 1: the <28 weeks gestational age row for England reads N/A for all categories. Should this be 0, or is there really no data?

7- Please break up the Supplementary information into individual files for each item.

Comments from Reviewers:

Reviewer #2: All of my comments were adequately addressed. 

Reviewer #3: The authors have done a very thorough review, I commend them.

I apologize that I am short on time and therefore I only go through my own comments (and my recommendation should be seen in this light).

My Q3: I meant antibiotics DURING birth. Interesting that the authors have done studies on prenatal antibiotics and offspring outcomes. In my group and other groups large registry studies with clever designs have debunked the causal association between maternal antibiotics and offspring infections back in 2014, PMID: 25066330 and 25432937 (therefore I have many reservations for the microbiome hypothesis).

My Q8: when you adjust for GA in 2 week categories I dont think you fully adjust, that's all. But I agree that such an adjustment, although finer in resolution, would not at all remove your findings.

My Q9: statified Cox is different than adjustment since it handles the non-proportional covariates (and takes advantage of the semiparametric model that is Cox). But I can see that you already have many analyses, however it would have been nice to see in the reviewer comments.

My Q11: Thank you for including the negative control

My Q15: You are correct, I apologize.

The discussion has been re-written and is very clear now and with less "fluff". Highly appreciated. 

Reviewer #5: I was asked to review the earlier statistician's comments and the authors responses.

Both the comments and the responses were well done and I now recommend publication

Peter Flom

[LINK]

---

## [Editor Report · Decision Letter 3]

16 Oct 2020

Dear Dr Miller, 

On behalf of my colleagues and the academic editor, Dr. Gordon C Smith, I am delighted to inform you that your manuscript entitled "Mode Of Birth And Risk Of Infection-Related Hospitalisation In Childhood: A Population Cohort Study Of 7.17 Million Births From Four High-Income Countries" (PMEDICINE-D-20-00075R3) has been accepted for publication in PLOS Medicine. 

PRODUCTION PROCESS

Before publication you will see the copyedited word document (within 5 business days) and a PDF proof shortly after that. The copyeditor will be in touch shortly before sending you the copyedited Word document. We will make some revisions at copyediting stage to conform to our general style, and for clarification. When you receive this version you should check and revise it very carefully, including figures, tables, references, and supporting information, because corrections at the next stage (proofs) will be strictly limited to (1) errors in author names or affiliations, (2) errors of scientific fact that would cause misunderstandings to readers, and (3) printer's (introduced) errors. Please return the copyedited file within 2 business days in order to ensure timely delivery of the PDF proof. 

If you are likely to be away when either this document or the proof is sent, please ensure we have contact information of a second person, as we will need you to respond quickly at each point. Given the disruptions resulting from the ongoing COVID-19 pandemic, there may be delays in the production process. We apologise in advance for any inconvenience caused and will do our best to minimize impact as far as possible.

PRESS

PROFILE INFORMATION

Thank you again for submitting the manuscript to PLOS Medicine. We look forward to publishing it. 

Best wishes, 

Thomas McBride, PhD

Senior Editor 

PLOS Medicine

plosmedicine.org